# Systemic Treatments for Advanced Small Bowel Adenocarcinoma: A Systematic Review

**DOI:** 10.3390/cancers14061502

**Published:** 2022-03-15

**Authors:** Paola Di Nardo, Silvio Ken Garattini, Elena Torrisi, Valentina Fanotto, Gianmaria Miolo, Angela Buonadonna, Fabio Puglisi

**Affiliations:** 1Department of Medical Oncology, CRO Aviano, National Cancer Institute, IRCCS, 33081 Aviano, Italy; gmiolo@cro.it (G.M.); abuonadonna@cro.it (A.B.); fabio.puglisi@cro.it (F.P.); 2Department of Oncology, ASUFC University Hospital of Udine, 33100 Udine, Italy; silvioken.garattini@asufc.sanita.fvg.it (S.K.G.); valentina.fanotto@asufc.sanita.fvg.it (V.F.); 3Department of Medical Oncology, P.O. San Vincenzo, 98039 Taormina, Italy; torrisi@oncologiataormina.it; 4Department of Medicine (DAME), University of Udine, 33100 Udine, Italy

**Keywords:** small bowel adenocarcinoma, clinical trials, chemotherapy, target therapy, immunotherapy, systematic review

## Abstract

**Simple Summary:**

Small bowel adenocarcinoma is a rare disease that is usually treated following the protocols in use for colorectal cancer. The aim of this systematic review was to identify the optimal treatment for this disease according to the currently available evidence. We concluded that there is some evidence regarding the use of chemotherapy doublets in this setting, but not to support the use of biological agents. There are encouraging results regarding the use of immunotherapy in selected patients.

**Abstract:**

Small bowel adenocarcinoma (SBA) is a rare disease for which scarce evidence is available. We summarized data available on systemic treatment of advanced SBA. Methods: Scientific literature was evaluated to find phase II or phase III clinical trials on systemic treatment for advanced SBA. MeSH terms were selected and combined for the initial search, then inclusion and exclusion criteria were set in a search protocol. Four medical oncologists looked for evidence on Medline, EMBASE and Cochrane databases. Moreover, abstracts from 2016 to June 2021 from the American Society for Clinical Oncology, European Society for Medical Oncology, Gastrointestinal Cancer Symposium and World Congress on Gastrointestinal Cancer were browsed. The selected studies, matching the inclusion and exclusion criteria, were finally tabulated and analyzed. Results: The trials finally selected were 18 phase II/III clinical trials. Four small phase II trials support the activity of oxaliplatin-based doublets in first-line treatment (CAPOX and mFOLFOX). Conclusion: No good level evidence is available on the use of bevacizumab, anti-epidermal growth factor receptor, targeted agents or immunotherapy. First-line treatments are largely derived from colorectal cancer protocols, mainly oxaliplatin-based doublets.

## 1. Introduction

Small bowel adenocarcinoma (SBA) is a rare tumor, and it accounts for 0.6% of the newly diagnosed cases of cancer in the United States every year [1]; however, it represents 36.9% of neoplasms arising from the small bowel, with an incidence only slightly lower than the one of neuroendocrine tumors [2]. Most commonly, SBA localizes to the duodenum (48–59%), jejunum (19–29%) and ileum (10–15%) [3,4]. Unfortunately, due to the paucity of cases there is a dearth of robust evidence for the treatment of this disease. SBA is a largely neglected type of cancer if compared with other neoplasms of this district, such as gastro-intestinal stromal tumors (GIST) or neuroendocrine tumors, for which specific treatments are available; on the contrary, the backbone of the treatment for this neoplasm is represented by surgery, and little is known on the optimal management of advanced disease. Even medical oncologists focusing on gastro-intestinal cancers have poor treatment options to face this disease, which is typically treated with chemotherapeutic agents commonly in use for metastatic colorectal cancer (mCRC). The majority of the studies on SBA are either retrospective or case series investigating combinations of chemotherapeutics, mainly doublets of fluoropyrimidines associated with oxaliplatin or irinotecan. Unlike colorectal cancer, very little is known on the efficacy of biologics in the treatment of advanced SBA; some reports describe the molecular landscape of this carcinoma, demonstrating that 40–60% of SBAs carry a KRAS mutation [5], similar to what is observed in mCRC, and thus suggesting a potential response to epidermal growth factor receptor (EGFR) inhibitors. SBA differs from mCRC as for the incidence of microsatellite instability (MSI), whose prevalence has been reported to range from 20% to 33% [6], hence suggesting a potential for the use of immune checkpoint inhibitors in the treatment of SBA. Despite these considerations, mCRC-like treatment without biologics represents the mainstay of the current management of this neoplasm. In this systematic review we evaluate the literature to find evidence supporting the use of chemotherapy in advanced SBA, appraising potential associations with biologic agents and identifying the most recent studies on new therapeutic approaches such as immunotherapy.

## 2. Methods

### 2.1. Search Strategy

We performed a literature search to identify studies on the impact of different systemic treatments in advanced small bowel disease. English language articles referring to clinical trials were looked for on EMBASE, Medline and the Cochrane database. In addition, we analyzed the on-line abstract books of the American Society for Clinical Oncology (ASCO), the European Society for Medical Oncology (ESMO), the Gastrointestinal Cancer Symposium (ASCO GI) and the World Congress on Gastrointestinal Cancer (WCGC) organized from 2016 to June 2021, with the aim of updating the search to the latest unpublished clinical trials. We performed our search combining the following MeSH terms: “small bowel adenocarcinoma”, “small bowel cancer”, “small bowel carcinoma”, “small bowel neoplasm”, “small intestine cancer”, “small intestine adenocarcinoma”, “small intestine carcinoma”, “small intestine tumor”, “small intestine neoplasm”, “treatment”, “management”, “biologic therapy”, “biologics”, “target”, “EGFR”, “bevacizumab”, “panitumumab” and “cetuximab”.

At publication, this review is being assessed for registration by the PROSPERO Editorial board (ID 314175; https://www.crd.york.ac.uk/prospero/, accessed on 2 March 2022).

### 2.2. Eligibility Criteria

The PICO format was used to define the selection criteria of relevant studies. 

Population: patients affected by unresectable or metastatic small bowel adenocarcinoma; any age, any gender and any performance status, enrolled in phase II/III clinical trials.Intervention: Any type of treatment for advanced unresectable disease, including chemotherapy combinations, biological drugs (such as anti-vascular endothelial growth factor receptor biologics, tyrosine kinase inhibitors, anti-EGFR monoclonal antibodies) and immunotherapy.Comparison: This criterion did not apply to the aims of this systematic review.Outcomes of interest: Efficacy of treatments for advanced small bowel adenocarcinoma, including overall survival, progression free survival and overall response rate.

### 2.3. Exclusion Criteria

Studies conducted on cancers of the small bowel other than SBA, such as neuro-endocrine tumors and GIST, were excluded. Phase I trials were excluded. 

### 2.4. Data Collection

The search was independently conducted with the above-mentioned criteria by four reviewers with expertise in the field of gastro-intestinal medical oncology. Each reviewer checked for elimination of duplicated articles. Finally, the papers and abstracts selected were evaluated by three other reviewers; they were matched and assembled in an Excel file for the final assessment and manuscript writing. Whenever more than one report was available for the same trial, the latest published results were considered.

## 3. Results

After the first round of search, we identified 197 articles matching the chosen keyword combinations. The list of articles consisted of 184 publications and 13 abstracts from the most recent congresses in the fields of oncology and gastro-intestinal oncology (ASCO, ESMO, ASCO GI and WCGC by ESMO). We selected a core of 64 articles after excluding duplicates (Figure 1). Subsequently, we eliminated the articles that did not match the inclusion and exclusion criteria set at the time of the search-protocol development and, whenever more than one article pertained the same study, we chose the most recent report. Following in-depth evaluation of the abstracts, 13 studies turned out to be retrospective, 4 studies fell within the review article category, 12 studies were phase I trials, 1 study was a case report, 1 study was a preclinical report and 1 study was a basket trial with only one patient affected by a small bowel adenocarcinoma. Of the remaining 32 articles, 6 pertained studies for which a more recent publication was available. Among a total of 26 phase II/III clinical trials, 6 studies were excluded, as they were carried out in the neo/adjuvant setting, and 2 studies were not considered, being radiotherapy trials. A total of 18 studies turned out to be fully eligible; they were tabulated in an Excel file and analyzed for their efficacy endpoints (Table 1).

### 3.1. Chemotherapy Combinations

In our systematic search we identified 7 clinical trials, including one pharmaco-genetic study on irinotecan. In this multicentric trial [7], 33 treatment-naïve SBA patients were divided in three cohorts of treatment based on the UGT1A1*28 genotype (UGT1A1*28 6/6, UGT1A1*28 6/7 and UGT1A1*28 7/7). Patients received a capecitabine, oxaliplatin and irinotecan (CAPIRINOX) combination; doses were based on the UGT1A1*28 genotype. The maximal dose was administered to patients with the genotype UGT1A1*28 6/6, while the minimal dose was given to patients with the UGT1A1*28 7/7 genotype. The genotype-based arms were pooled for toxicity and efficacy analyses. A total of 12 out of the 32 patients analyzed (38%; 95% CI: 21–56%) achieved a response; toxicity did not significantly differ among the different genotype-based doses. The median overall survival (mOS) attained by this regimen was 13.4 mo (95% CI: 21–56%); the mPFS was 8.9 mo (95% CI 4.7–10.8 months). The authors concluded that genotype-based dosing of this treatment determines a prolonged response that needs to be further compared with a capecitabine and oxaliplatin (CAPOX)-based treatment.

In a phase II clinical trial published in 2005, a combination of 5-fluoruracil, doxorubicin and mitomycin C (FAM regimen) was tested in advanced SBA [8]. This was a multi-institutional study that included pretreated patients and whose primary endpoint was the overall response rate (ORR). A total of 36 of the 39 patients included in the study were evaluable, and the ORR determined was 18.4%, with a mOS of 8 mo. This result was judged by the authors to be insufficient to claim superiority of the FAM combination over other therapeutic regimens. The CAPOX regimen was tested on patients with previously untreated SBA or carcinoma of the ampulla of Vater in a phase II trial organized by Overman and colleagues in 2009 [9]. The chemotherapeutic regimen was oxaliplatin administered on day 1 at the dose of 130 mg/m^2^ associated with capecitabine 750 mg/m^2^ bis-in-die for 14 days on a 21-day cycle. The primary endpoint was ORR. The 31 patients of the trial provided encouraging results with a 50% ORR (3 complete responses), a median time to progression (mTTP) of 9.4 mo (95% CI: 4.4– > 35) and a mOS of 20.4 mo (95% CI: 14.4– > 35). Hence, it was stated that CAPOX should be regarded as a first-line standard treatment in this neoplasm. Very similar conclusions were drawn in 2013 by Kim et al. [10]. This group used the same regimen on a cohort of 21 patients affected by Vater’s ampulla intestinal-type cancers. A total of 18/21 patients underwent first-line treatment with CAPOX, while 3 patients were subjected to the same regimen as second-line treatment. The efficacy outcomes measured in the trial were mTTP and mOS. Consistent with what was observed by Overman et al. [9], the calculated mTTP value was 7.6 mo (95% CI: 6.7–8.5), and the mOS value was 19.7 mo (95% CI: 14.8–23.6). In a Japanese phase II trial published in 2017 [11], the mFOLFOX6 regimen (oxaliplatin 85 mg/m^2^ day 1, leucovorin 200 mg/m^2^ day 1, fluorouracil bolus 400 mg/m^2^ day 1 and 5-fluorouracil continuous infusion 2400 mg/m^2^ over 46 h every 14 days) was administered to 24 patients. In this first-line, single-arm, open-label phase II trial, mFOLFOX showed a 1-year progression-free survival (mPFS) in 23.3% of the patients (mPFS 5.9 mo, 95% CI: 3.0–10.2, primary endpoint) and a 17.3 months mOS (95% CI: 11.7–19.0), with a 45% ORR. Moreover, the FOLFOX regimen was used as first-line treatment in a phase II clinical trial (33 patients) by Xiang et al. [12], which resulted in a 48.5% ORR value (95% CI: 31–67%), a 7.8 mo mTTP (95% CI: 6.0–9.6) and a 15.2 mo mOS (95% CI: 11–19.4). The most recent phase II study evaluating the effects of chemotherapy in SBA was published by Overman et al. [13] in 2018. Patients with a CpG island methylator phenotype (CIMP) mCRC and patients with SBA refractory to first-line treatment were treated with nab-paclitaxel 260 mg/m^2^ for 1 day every 3 weeks. The rationale of this study relied on the evidence that cytosine methylation of the CpG islands within gene-promoters induces an inactivation of tumor suppressor genes. Tumors that share this type of promoter hyper-methylation are defined as CIMP-high. Preclinical studies provided evidence that taxanes show activity in CIMP-high cancers and colorectal tumors characterized by lack of mutations in the adenomatous polyposis coli (APC) gene, two related types of neoplasia. As SBAs are often characterized by the lack of APC gene alterations, these tumors were included in the study. Hence, the therapeutic protocol was applied to a cohort of 21 patients, which included 10 cases of SBA. In these patients, 2 partial responses were observed, which led the authors to conclude that nab-paclitaxel may represent a novel therapeutic option in advanced SBA.

### 3.2. Targeted Therapy

There is limited scientific literature on the use of combinations of chemotherapy and biological agents in SBA; the data available are generally restricted to case reports. Only two phase II clinical trials have been reported: (1) a small phase II, single-arm study evaluating the efficacy of a combination based on bevacizumab (a humanized anti-VEGF monoclonal antibody) and chemotherapy; (2) a single-center, single arm, Bayesian phase II trial on panitumumab (a fully human anti-EGFR monoclonal antibody).

#### 3.2.1. Anti-Vascular Treatment

An open-label, single arm, single-institution, phase II study was conducted to evaluate the benefit of adding bevacizumab to CAPOX chemotherapy in SBA and ampullary adenocarcinoma (AAC) [14]. Treatment consisted of oxaliplatin (130 mg/m^2^) on day 1, bevacizumab (7.5 mg/kg) on day 1 and oral capecitabine (1500 mg/m^2^) on days 1–14 of a 21-day cycle. Treatment was continued until disease progression or inacceptable toxicity. A total of 23 patients (77%) with SBA and 7 patients (23%) with AAC were enrolled. The primary endpoint of the study was mPFS, which showed an 8.7 mo value (95% CI: 4.9–10.5; 95% CI: 6.5–10.3) at a median follow-up of 25.9 mo. The secondary endpoint was median OS, whose calculated value was 12.9 mo (95% CI: 9.2–19.7; 90% CI: 10.5–17.2). The probability of PFS at 6 months was 68% (95% CI: 52–88%). The treatment induced ORR in 43.8% of the patients, and it resulted in 1 complete remission and 13 partial remissions. Despite the presence of intact small bowel primary tumors in 60% of the patients, there were no episodes of bowel perforation. The RR was similar in SBA (50%) and AAC (43%) patients.

An exploratory analysis comparing the data obtained in this study with the results of a previous phase II clinical trial [9] conducted by the same institution in SBA and AAC patients treated with CAPOX alone demonstrated no significant difference in RR (48.3% vs. 52%, *p* = 0.79) or mPFS (8.7 mo vs. 6.6 mo; *p* = 0.73; hazard ratio 1.125, 95% CI: 0.585–2.163). However, it must be emphasized that both studies were based on a small number of patients and were not designed or powered to detect differences between the two regimens.

An ongoing trial is currently exploring the combination of ramucirumab and paclitaxel versus the chemotherapy protocol FOLFIRI, following progression on a previous line of treatment with either a fluoropyrimidine, oxaliplatin or both; however, results have not been reported yet [15].

#### 3.2.2. Anti-EGFR Treatment

The efficacy of panitumumab monotherapy in refractory metastatic RAS wild-type SBA and AAC was evaluated in a single-center, open-label, single arm, Bayesian phase II trial [16]. Panitumumab was administered at a dose of 6 mg/kg intravenously every 14 days. The primary endpoint of the study was RR. The study was discontinued early, when only nine patients (one AAC case and eight SBA cases) were enrolled. While no patient showed mutations in KRAS or NRAS, two patients had a BRAF G469A mutation, and one patient was characterized by a PIK3CA H1047R mutation. Panitumumab did not induce any objective response. Indeed, two patients had stable disease, and seven of them progressed. At a median follow-up time of 16.6 mo, the median PFS value was 2.4 mo (95% CI: 1.5-NA), and the mOS value was 5.7 mo (95% CI: 2.7-NA). The treatment was well tolerated and resulted in the expected type of toxicity.

#### 3.2.3. Anti-BRAF Treatment

The ROAR study is an international, multicenter, open-label phase II trial (NCT02034110) that uses a Bayesian hierarchical statistical design. The ROAR trial evaluates the combination of dabrafenib (RAF-inhibitor) and trametinib (MEK-inhibitor) in subjects with rare cancers, including SBA cases characterized by a BRAF V600E mutation, an advanced stage and no standard treatment options [17]. The primary endpoint of the study is ORR, while the secondary objectives include the median duration of the response (mDOR), mPFS, mOS and safety. Pharmacodynamic markers and quality of life are other parameters that will be evaluated. The study is completed, but results for the small bowel cohort have yet to be published.

#### 3.2.4. Other Targeted Agents

Treatment of six subjects with advanced refractory SBA based on the identified alteration of human epidermal growth factor receptor 2 (HER2), EGFR and Hh pathway with standard doses of trastuzumab and pertuzumab, erlotinib or vismodegib, respectively, within an open-label phase IIa basket trial (MyPathways) did not induce any response [18].

### 3.3. Immunotherapy

There is a dearth of studies focusing on immunotherapy in small bowel adenocarcinomas. Four phase II clinical trials investigating the role of immunotherapy in advanced SBA are available. In addition, the results of two immunotherapy-based, single-arm, phase II trials were recently reported at major international conferences.

The French AcSé immunotherapy program allowed access to anti-PD1 therapies outside their current approval. In this program, a phase II trial [19] was conducted to evaluate nivolumab in rare cancers. Nivolumab (240 mg IV) was administered q2w. A total of 50 patients were enrolled in this trial; of these, 7 patients had a small bowel adenocarcinoma. ORR at 12 weeks was 38% (95% CI: 24.6% to 52.8%); DCR was 74%. Median PFS was not reached with a 6 mo PFS at 58.9% (95% CI, 46.5% to 74.6%). At the date of analysis, the 6 mo OS rate was 80.3%.

The Keynote 158 study evaluated the efficacy of programmed cell death protein 1 (PD-1) blockade in the same set of patients described above. In this trial, pembrolizumab was administered to patients affected by MSI-H non-colorectal cancer who had failed at least one prior line of therapy. Eligible patients received 200 mg pembrolizumab every 3 weeks until progression, unacceptable toxicity or patient’s/physician’s decision. The study enrolled 223 patients, including 19 cases of small intestinal cancer. The primary endpoint of the study was ORR, and the secondary endpoints included mDOR, mPFS, mOS and safety. ORR was observed in 42.1% of the patients affected by SBA (95% CI: 20.3–66.5%). The median PFS was 9.2 months, while the median OS was not reached at the time of the report (January 2020). The observed adverse events (AEs) were in line with the ones already reported for pembrolizumab; 14.6% of all patients in the study had serious drug-related AEs [20,25,26].

A phase II basket trial evaluated the activity of pembrolizumab in dMMR cancers, independent of tumor histology. The trial enrolled patients who had failed at least one prior line of treatment. Pembrolizumab was administered at 10 mg/kg every 14 days. The co-primary endpoints of the study were the ORR and PFS rates at 20 weeks. The secondary endpoints included the disease control rate (DCR), mPFS, mOS and safety. A total of 46 non-colorectal patients were enrolled, including 5 small bowel cancer patients. The ORR and DCR values were 54% (95% CI: 39–69%) and 72% (95% CI: 57–84%), respectively. Median PFS was 18.1 months. The median OS was not reached at the time of the last analysis, but the 2-year OS was 57% [21,27,28].

A multicenter, single-arm phase II clinical trial aimed at establishing the activity of pembrolizumab (200 mg IV every 3 weeks) in patients with unresectable or metastatic SBA refractory to first-line chemotherapy. The trial enrolled 40 patients irrespective of the tumor mismatch-repair status. The primary endpoint of the study was the response rate, which did not meet predefined success criteria of ORR 30% (3/40 PR, 8%; 95% CI: 2–20%); of these three patients, one had a low MSS/MSI status but a high tumor mutation burden. Median OS was 7.1 months (95% CI, 5.1–17.1) and median PFS 2.8 months (95% CI, 2.7–4.2. Fifty percent of patients with MSI-H tumors achieved PR and remained alive without progression. Twenty-five patients (63%) had serious adverse events (at least grade 3) [22].

A pilot study to define safety and efficacy of avelumab in small bowel adenocarcinomas was recently reported at the ASCO GI conference in 2020. A total of 8 SBA patients (5 small intestine; 3 ampullary) were enrolled; of these, 7 were evaluable for efficacy, with a median time on treatment of 3.4 months. ORR was 29% (3/7 pts), while DCR was 71% (5/7). Median PFS was 8.0 months [23].

The DART/SWOG S1609 aimed to test the combination of anti-CTLA-4 and anti-PD-1 blockade with ipilimumab and nivolumab in rare tumors. A total of 25 patients were enrolled in the small bowel cohort; the overall response rate was 8%, and the median PFS was 2 months; 6-month OS was 48%, and median OS was 6 months [24].

## 4. Discussion

Advanced SBA is a rare disease that is generally managed according to the treatment paradigms established for mCRC. In clinical practice, the use of chemotherapeutic regimens involving combinations between fluoropyrimidines and either oxaliplatin or irinotecan is predominantly based on expert opinions and case reports rather than on solid evidence. Nevertheless, the scientific literature contains clinical trials, which can help to guide the decision making of medical oncologists. In this systematic review we identify seven phase II studies addressing the question as to which chemotherapy regimen is to be used in advanced SBA. Unfortunately, all the identified trials were small, having a median sample size which ranged from 20 to 30 patients. This emphasizes the difficulty in organizing multicenter studies, due to the rarity of advanced SBA. The results obtained in the aforementioned trials support the use of oxaliplatin and fluoropyrimidine-based regimens for the treatment of this disease [9,10,11,12,13]. Our analysis indicates a good external consistency for the efficacy outcomes obtained in the trials based on mFOLFOX or CAPOX regimens. In the reported studies, the median ORR ranged from approximately 40% to 50%, while the mOS reached values of 17–20 months, which is likely to reflect the real impact of these chemotherapy schemes on the evolution of SBA. Thus, for SBA there is still a need of looking for good level evidence about which chemotherapy is the most effective even in a first-line setting. In addition, our systematic review indicates that there is no significant published evidence supporting the use of irinotecan-based regimens in the treatment of SBA. The only study on irinotecan is a genotype-based phase II trial involving the administration of the CAPIRINOX triplet [7]. Although the treatment proved to be active (38% ORR), the mOS value reported is in line with the historical counterpart derived from oxaliplatin doublets. The addition of anthracyclines and mitomycin to fluorouracil did not turn into a clinically active combination [8]. Given this lack of convincing trials in favor of a first-line doublet standard, it may be useful to concentrate research efforts towards the optimal administration of first- and second-line doublets, rather than treatment intensification.

In line with the current mCRC treatments, there is a report focusing on the addition of bevacizumab to chemotherapy for the treatment of advanced SBA [14]. Unfortunately, this phase II non-comparative study shows no added value of the bevacizumab-based combination when the results are compared with the historical data available on oxaliplatin doublets. On the other hand, the observation that 40–60% of SBA carries a KRAS mutation led to the design of trials aimed at blocking the EGFR-dependent proliferation pathway. Once again, only one small phase II trial is available on the issue [16]. The study tested panitumumab as a single agent and it showed a lack of efficacy in pretreated advanced SBA. The other evidence available is based on case series and retrospective studies on bevacizumab, cetuximab and panitumumab in combination with chemotherapy. These reports provide heterogeneous results that need to be interpreted accurately. As observed in the case of mCRC, some improvement in the appropriate selection of patients is required. This must be based on the results of translational research aimed at the identification of factors predicting the efficacy of biologics and immunotherapy. At present, extended RAS analysis has proved ineffective for the selection of patients responding to EGFR inhibitors. Finally, our analysis demonstrates that there are a few completed or ongoing basket trials based on treatments targeting specific alterations such as BRAF V600E or HER2 mutations/amplifications [17,18].

A promising research strategy relates to the observation that up to 35% of the SBA cases present with MSI. However, the currently available studies are predominantly “proof of concept” basket trials, which include few SBA patients, rather than convincing reports. In the field of immunotherapy, the results of the available studies are promising, but they are not yet compelling due to the small number of patients involved [19,20,21,22,23,24,25,26,27,28]; however, in the near future, due to the scarcity of options, we expect that the MSI status or defective mismatch repair will guide the selection and treatment of SBA patients with immune checkpoint inhibitors.

Our systematic review highlights the absence of reliable evidence supporting the use of anticancer regimens other than oxaliplatin-based doublets for the treatment of advanced SBA. The lack of phase III randomized clinical trials makes it difficult to draw definitive conclusions as to the proper sequence of treatments to be implemented following first-line management of SBA patients. It cannot be excluded that the results of some clinical trials may have escaped our analysis, as some are still unpublished or were not identified by the keywords used in our study. In fact, the inclusion of abstracts lacking supplementary information may have resulted in insufficient details on the characteristics and the results of specific trials. However, we feel that the inclusion of congress abstracts is necessary for providing an updated review that is inclusive of the latest research data. With respect to these last considerations, it should be remarked that a formal meta-analysis of the available clinical trials on advanced SBA is difficult and unlikely to show meaningful results, given the lack of homogeneity of treatments, endpoints and results.

## 5. Conclusions

The studies on the systemic treatment of advanced SBA are still scarce and mostly have a retrospective design. The most solid data derive from small phase II clinical trials supporting the use of FOLFOX or XELOX regimens as standards for the first-line treatment of this disease. Overall, the evidence available is insufficient to support the addition of targeted agents to chemotherapy or the use of immunotherapy in the treatment of advanced SBA, although the results of the aforementioned immunotherapy trials are encouraging.

## Figures and Tables

**Figure 1 cancers-14-01502-f001:**
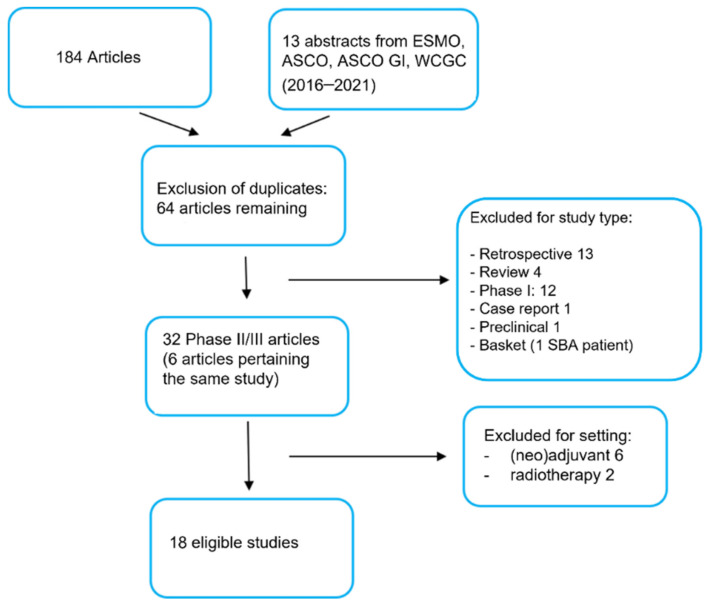
PRISMA chart. Description of the process of selection of the studies analyzed for the systematic review. The selected keywords generated a first list of full studies and abstract (197). Then, inclusion and exclusion criteria were applied. Finally, eighteen studies were selected, tabulated and analyzed for the final evaluation.

**Table 1 cancers-14-01502-t001:** Studies regarding systemic treatments in advanced small bowel adenocarcinoma.

	Year	N° pts	Study Type	Treatment	Setting	Primary Outcome	Other Outcomes	Reference
Chemotherapy								
McWilliams R.R. et al.	2017	33	phase II	CAPIRINOX (dose on UGT1A1*28 genotype)	I line	ORR 37.5% (IC 95% n.r)	mOS 13.4 m mPFS 8.9 m	[7]
Gibson M.K. et al.	2005	39	phase II	Fluorouracil + Mitomycine C + Doxorubicine	I line/pretreated	ORR 18% (IC 95% 7.8–34.4)	mOS 8 m	[8]
Overman M.J. et al.	2009	31	phase II	CAPOX	I line	ORR 50% (IC 95% 31–69)	mTTP 9.4 m/ mOS 20.4 m	[9]
Kim H.S. et al.	2013	21	phase II	XELOX	I line	TTP 7.6 m (IC 95% 6.7–8.5)	mOS 19.7 m/ ORR 38% (2 CR)	[10]
Horimatsu T. et al.	2017	24	phase II	mFOLFOX	I line	1y PFS 23.3% (IC 95% 8.6–44.2%)	mOS 17.3 m/ ORR 45%/ mPFS 5.9 m	[11]
Xiang X.J. et al.	2012	33	phase II	mFOLFOX	I line	ORR 48.5% (95% CI 31–67%)	mTTP 7.8 m/ mOS 15.2 m	[12]
Overman M.J. et al.	2018	10	phase II	Nab-paclitaxel	II line or later line	2 PR (10 pts)	//	[13]
**Biologics**								
Gulathi P. et al.	2017	30	phase II	CAPOX + Bevacizumab	I line	PFS 8.7 m	mOS 12.9 m/ORR 43–50%	[14]
Overman et al.	2019	ongoing	phase II	Paclitaxel/Ramucirumab vs FOLFIRI	II line	PFS	OS, ORR, safety, CEA	[15]
Gulathi P. et al.	2018	9	phase II	Panitumumab single agent	pretreated	early stopped after 9 pts RR 0%	mPFS 2.4 m/mOS 5.7 m	[16]
Subbiah V. et al.	2016	ongoing	phase II basket	Dabrafenib + Trametinib for BRAF V600 mutant	pretreated	ORR	DOR, PFS, OS, safety	[17]
Hainsworth J.D.	2018	ongoing	phase IIa basket	Pertuzumab plus Trastuzumab, Erlotinib, Vemurafenib or Vismodegib (target-based)	pretreated	ORR 0%		[18]
**Immunotherapy**								
Tournigand C.	2019	50 MSI-H (7 SBA)	phase II	Nivolumab	All lines	ORR 38% (95% CI: 24.6% to 52.8%)	DCR 74%. 6 mo mPFS 58.9%	[19]
Marabelle A.	2020	233 MSI-H non-CRC (19 SBA)	phase II	Pembrolizumab	pretreated	ORR 42.1% (95% CI: 20.3–66.5%).	mPFS 9.2 mo. mOS not reached 14.6% SAE	[20]
Dung T. Le	2017	46 MMRd non-CRC (5 SBA)	phase II	Pembrolizumab	pretreated	ORR 54% (95% CI: 39–69%)	DCR 72% (95% CI: 57–84%) mPFS 18.1 mo 2 y OS 57%	[21]
Pedersen K.	2021	MSI tumors	phase II	Pembrolizumab	pretreated	ORR 8%	mOS 7.1 mo mPFS 2.8 mo sAE > G2 63%	[22]
Cardin D. et al.	2020	8	phase II	Avelumab	All lines	ORR 29%	DCR 71% mPFS 8.0 mo	[23]
Patel S.P. et al.	2020	25	phase II basket	Ipilimumab/Nivolumab	pretreated	ORR 8%	mPFS 2 mo 6 mo OS 48% mOS 6 mo	[24]

Tabular view of the 18 studies included in the final systematic review organized per type of treatment. SBA = small bowel adenocarcinoma, CRC = colorectal cancer, MSI-H = microsatellite instability high, MMRd = mismatch-repair deficient.

## Data Availability

All the material used for the purpose of this review is included in the manuscript; the referenced articles are available on public scientific databases.

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
