# Peer review of "Systemic Treatments for Advanced Small Bowel Adenocarcinoma: A Systematic Review"

_cancers, 2022, doi:10.3390/cancers14061502_

Round 1

Reviewer 1 Report

Review for cancers-1617911

In this article entitled “Systemic treatment of advanced small bowel adenocarcinoma: A systematic review”, the authors evaluated the scientific literature to find phase II/III clinical trials on systemic treatment of small bowel adenocarcinoma (SBA). A total number of 18 published studies were analyzed according to the inclusion and exclusion criteria. Taking into account the rarity of the disease, the few available treatment options, and the limited number of cases in the previously published reports, it was deduced that immunotherapy seemed better than chemotherapy and other biologics. Nevertheless, the literature scanning did not show good level of evidences regarding the different applied treatments.

Hereafter some comments revealed after reviewing the paper.

  • I guess it would be better to add an “s” to the word “treatment” in the title. As finally it was concluded that there is no relevant treatment, just encouraging results.
  • There is no title for the table 1, just footnotes.
  • References, in the body of the manuscript and particularly in the table should be checked. In fact, i) a consistent order is missing and ii) some reference numbers are not there (19, 20 and 21).
  • What does the superscript of 1 represent in the table footnote? Shouldn’t be deleted?
  • In line 121, it is written “… 6 phase II clinical…” but the reported data exhibit that all the seven identified clinical trial were associated with phase II including the pharmacogenetic study of
  • McWilliam et al. (2017). Please rephrase this part, to make it clearer to the readers.
  • Punctuation, throughout the manuscript, should be revised together with English language. What’s “akin”, in line 53 ?

Author Response

We thank the reviewer for their comments. Below a point-by-point response to the reviewer's comments.

Hereafter some comments revealed after reviewing the paper.

  • I guess it would be better to add an “s” to the word “treatment” in the title. As finally it was concluded that there is no relevant treatment, just encouraging results.
    • The title has been changed as suggested.
  • There is no title for the table 1, just footnotes.
    • A title has been added as suggested
  • References, in the body of the manuscript and particularly in the table should be checked. In fact, i) a consistent order is missing and ii) some reference numbers are not there (19, 20 and 21).
    • A) References have been modified both in the body and in the table as to follow a more precise order.
    • B) References 19, 20 and 21 were not in the table as they are abstracts; the full papers they refer to are in the table. Abstract 19 (now renumbered as 21) refers to the study Keynote 158, referenced as number 23 (now 22). Reference 20 and 21 (now 23 and 24) are abstracts for the full study referenced at number 24 (now 25).
  • What does the superscript of 1 represent in the table footnote? Shouldn’t be deleted?
    • It has been deleted
  • In line 121, it is written “… 6 phase II clinical…” but the reported data exhibit that all the seven identified clinical trial were associated with phase II including the pharmacogenetic study of
    • It has been changed to be more accurate
  • McWilliam et al. (2017). Please rephrase this part, to make it clearer to the readers.
    • It has been rephrased to be more clear.
  • Punctuation, throughout the manuscript, should be revised together with English language. What’s “akin”, in line 53 ?
    • “Akin” has been changed to “similarly”. Some changes have been made to punctuation and English.

Reviewer 2 Report

The PICO (patient, intervention, comparator, outcome, and study design) format should be used  as the strategy  to facilitate a systematic review process to avoid bias in the selection of studies.

There are several outcomes for an meta-analysis, why is this not executed?

Please contact a  methodology consultant 

Author Response

We thank the reviewer for their comments. Here is our point-by-point response.

The PICO (patient, intervention, comparator, outcome, and study design) format should be used  as the strategy  to facilitate a systematic review process to avoid bias in the selection of studies.

  • We used the PICO format when we defined the review protocol, but it was not clarified in the methods. This has now been amended and is, hopefully, more clear.

There are several outcomes for an meta-analysis, why is this not executed?

  • There is a conspicuous lack of homogeneity of treatments, endpoints and results among the reported trials; as such, we felt that a formal metanalysis on these data would not allow any meaningful conclusion. This is clarified at the end of the discussion.

Round 2

Reviewer 2 Report

You have a good material but please contact a specialist in the field of meta-analysis for advice.

I am not sure your are using principles of PICO and GRADE criteria correct.

Author Response

You have a good material but please contact a specialist in the field of meta-analysis for advice.

  • We thank reviewer 2 for this comment. The scope of this review was to analyse and summarise the available evidence regarding systemic treatments for advanced small bowel adenocarcinoma. While analysing the available studies, we realised that there was a conspicuous lack of homogeneity among them; as such, we decided not to perform a metanalysis. However, we will heed this advice for future studies when more homogeneous data regarding this subject will be available.

I am not sure your are using principles of PICO and GRADE criteria correct.

  • We thank the reviewer for this comment. As we could not perform a metanalysis, the GRADE approach – which is used to assess evidence or develop recommendations, with the aim of providing guidelines – was not considered appropriate as it was outside the purpose of our review.

About PICO, we reviewed it; we confirm that our criteria are as follows:

  • Population: patients affected by unresectable or metastatic small bowel adenocarcinoma; any age, any gender and any performance status, enrolled in phase II/III clinical trials.
  • Intervention: Any type of treatment for advanced unresectable disease, including chemotherapy combinations, biological drugs (such as anti-vascular endothelial growth factor receptor biologics, tyrosine kinase inhibitors, anti-EGFR mono-clonal antibodies) and immunotherapy.
  • Comparison: it did not apply to the aims of this systematic review.
  • Outcomes of interest: Efficacy of treatments for advanced small bowel adenocarcinoma, including overall survival, progression free survival and overall response rate.

This manuscript is a resubmission of an earlier submission. The following is a list of the peer review reports and author responses from that submission.